# FedRule: Federated Rule Recommendation System with Graph Neural Networks

**Yuhang Yao** *
Electrical and Computer Engineering
Carnegie Mellon University
yuhangya@andrew.cmu.edu

**Mohammad Mahdi Kamani**
AI Team
Wyze Labs
mkamani@wyze.com

**Zhongwei Cheng**
AI Team
Wyze Labs
zcheng@wyze.com

**Lin Chen**
AI Team
Wyze Labs
lchen@wyze.com

**Carlee Joe-Wong**
Electrical and Computer Engineering
Carnegie Mellon University
cjoewong@andrew.cmu.edu

**Tianqiang Liu**
AI Team
Wyze Labs
tliu@wyze.com

## Abstract

Much of the value that IoT (Internet-of-Things) devices bring to "smart" homes lies in their ability to automatically trigger other devices' actions: for example, a smart camera triggering a smart lock to unlock a door. Manually setting up these rules for smart devices or applications, however, is time-consuming and inefficient. Rule recommendation systems can automatically suggest rules for users by learning which rules are popular based on those previously deployed (e.g., in others' smart homes). Conventional recommendation formulations require a central server to record the rules used in many users' homes, which compromises their privacy and leaves them vulnerable to attacks on the central server's database of rules. Moreover, these solutions typically leverage generic user-item matrix methods that do not fully exploit the structure of the rule recommendation problem. In this paper, we propose a new rule recommendation system, dubbed as `FedRule`, to address these challenges. One graph is constructed per user upon the rules s/he is using, and the rule recommendation is formulated as a link prediction task in these graphs. This formulation enables us to design a federated training algorithm that is able to keep users' data private. Extensive experiments corroborate our claims by demonstrating that `FedRule` has comparable performance as the centralized setting and outperforms conventional solutions.

## 1 Introduction

With the rapid expansion of smart devices and applications in recent years, it becomes imperative to automate the actions of different devices and applications by connecting them together. For example, an occupancy sensor change can trigger a smart thermostat to turn on, or a code merge can trigger software updates. These connections are broadly construed as *rules* between entities in different systems. Setting up these rules between many entities can involve a tedious and challenging search process, especially for new users. Hence, it is helpful to provide new users with meaningful recommendations by learning from other users' sets of rules.

Recommendation systems are growing in various applications from e-commerce [13] to social networks [19] and entertainment industries [1, 8]. In most cases, the recommendation problem is

---

*This work was done during Yuhang's internship at Wyze Labs.

Workshop on Federated Learning: Recent Advances and New Challenges, in Conjunction with NeurIPS 2022 (FL-NeurIPS'22). This workshop does not have official proceedings and this paper is non-archival.

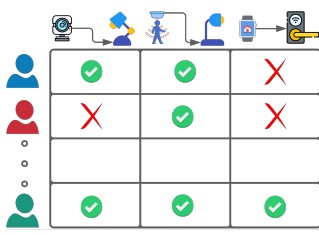
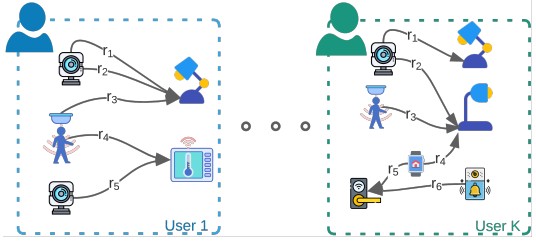

(a) User-Item Matrix Design         (b) Graph Structure Design

Figure 1: Illustrations of the conventional structure of recommendation systems based on user-item structure (left), and the proposed graph-based structure for rule recommendation system in a smart home device connections setting (right). In the conventional setting, each rule is considered as an item while in the graph structure each rule is represented as an edge between pairs of entity nodes.

formulated as a matrix completion task of a user-item matrix [6, 21, 2] to recommend new items to the users based on previous items chosen by the user. In this way, the recommendation problem is reduced to a dual link between users and items, as depicted in Figure 1(a). Following this setting, we consider each rule as an item for recommending them to users. However, this is not desirable as the structure of entities (*e.g.* the structure of devices or applications owned by a user) on the user side is not considered. Hence, we are not able to distinguish between different instances of an entity type (*e.g.* multiple cameras or multiple motion sensors) to provide meaningful recommendations for each separate entity based on their existing structure. Moreover, *privacy* and *security* concerns for training models on users' data in recommendation systems formulated as user-item matrix completion become a great challenge in IoT systems. The user-item formulations require a central server to know all users' rules: these may be sensitive, especially if they involve smart devices' behavior in users' homes.

In this paper, we, first, propose a new rule recommendation framework dubbed as `GraphRule`, based on the graph structure of entities for each user. In our proposed setup, we create a directed graph for each user based on their available entities (*e.g.* light bulb and contact sensor) as nodes and their rule connections as edges in this graph. Therefore, instead of representing a user's rules as a row in the user-item matrix (each item is an entity-rule-entity triplet, *e.g.* when contact sensor is open, turn on light bulb), we represent each users' rules as a graph that encapsulates the structure of entities and how they are connected together through the available rules. As an example, in Figure 1(b), for user 1, we have two separate cameras denoted by different nodes in the graph, each connecting to a distinct set of devices by their specific rules. However, it is infeasible to distinguish between these cameras in the user-item structure. In fact, in user-item design, we condense entities with the same type into one general entity (*e.g.* camera 1 and camera 2 into camera) to only emphasize on their connections to other entities as rules. Based on our graph structure design, the goal of the rule recommendation system is then to predict the newly formed edges (rules) in the graph, which can be formulated as a graph link prediction problem [22]. The system first learns node embeddings for each entity in the user's graph using a graph neural network [16, 18]. Based on the embeddings, a prediction model is then used to estimate the probability of different rules connecting each pair of entities in the graph.

For privacy concerns, unlike the conventional user-item structure, the proposed graph structure can be easily distributed across different users for training the main recommendation model locally. Indeed, by casting the rule recommendation problem as a link prediction task across graphs for different users, we can employ privacy-preserving training setups such as federated learning [14], which allows data to remain at the user's side. Moreover, federated training methods are generally iterative, and can thus easily adapt the model to new users' data instead of having to retrain the whole model from scratch, as is common in previous approaches for recommendation systems.

There has been a surge in applications of federated learning in the training of machine learning models recently [14, 12, 4, 9, 7], with some applications in recommendation systems [20, 5]. However, most of the proposals for recommendation tasks in federated learning are for *cross-silo* setups (*i.e.*, data could be distributed across pre-defined silos) based on user-item matrix completion, and not for *cross-device* settings [12] (*i.e.*, with limited data at each user, where here "device" represents one user and its entities), as in our rule recommendation task.

Our cross-device setting with the proposed graph structure, however, introduces new challenges: the amount of data on each individual user is very limited (as each user generally has only a few entities, e.g., a few smart home devices) and the local data is not independent and identically distributed (non-IID). The non-IID data distribution with a small sample size severely increases the variance of gradients among clients. Hence, when simply applying federated training methods, like FedAvg [14, 11], the model will not converge easily and cannot have comparable performance on par with the centralized training. To overcome the issue of severely non-IID data distribution across clients, we use two control parameters in local machines for each client to correct the gradients for different parts of our model and avoid drifting too much from the average model. With these solutions, we then introduce `FedRule`, a federated rule recommendation system with graph neural network. `FedRule` can learn the representation and link prediction models of the recommendation system at the user level with a decent convergence rate using variance reduction techniques with control parameters, while preserving the privacy of users' data. Our main contributions are:

- We propose a new rule recommendation framework, called `GraphRule`, based on the graph structure of current users by representing rules as edges between entity nodes over conventional user-item structure.
- We propose an effective federated learning system, called `FedRule`, which effectively learns from limited non-IID data on each user's rules with fast convergence and preserving the privacy of users' data.
- Extensive empirical investigation shows the effectiveness of both the proposed structure for the `GraphRule` and the `FedRule` algorithms. `FedRule` is able to achieve the same performance as the centralized version (`GraphRule`) due to its variance reduction mechanism.

## 2 Federated Rule Recommendation

### 2.1 Graph Rule Recommendation

In common recommendation systems, the goal is to predict unknown links between users and items. However, in rule recommendation problems we want to suggest how users can connect two or more entities (*e.g.* smart devices). In previous approaches, we are limited to considering multiple entities' connections as the connections of one abstract entity to create a user-item matrix (with items as rules). In this case, the item is a rule between those entities, and hence, we are losing information on the graph structure of entities and their connections.

In the rule recommendation problem, similar to the graph link prediction problem [22], we need to consider the relationship between different entities as well. Hence, instead of representing the connections as a row in a 2D matrix of user-item relationships, we have a graph of entities' connections for each user (see Figure 1). The graph also enables us to represent entities with the same type (*i.e.*, Cameras) as separate entity nodes in the graph. In addition, in this setting, compared to a simple graph, instead of binary connections between different entities, we can have multiple types of edge connections. This will enable us to represent different types of connections as different rules between entities. Using this structure, we can leverage advancements in the graph neural network domain to learn a better model for recommending new rules to the users.

Therefore, given the set of users $\mathcal{U}$ and the set of connection types $\mathcal{R}$, for each user $u_k \in \mathcal{U}$ in our setting, we create a graph of entities denoted by $\mathcal{G}_k = (\mathcal{V}_k, \mathcal{X}_k, \mathcal{E}_k)$, where $\mathcal{V}_k$ is a set of nodes (entities), $\mathcal{X}_k$ is the set of nodes' (entities') features (*e.g.*, types of entities) , and $\mathcal{E}_k$ is the set of connections between nodes in this graph. Each connection in $\mathcal{E}_k$ can be represented as $(v_i, v_j, r)$, which means there is an edge (rule) between node $v_i$ and node $v_j$ with connection type $r \in \mathcal{R}$ (e.g. "is open, turn on" in "when contact sensor is open turn on light"). The goal of the recommendation system in this problem is to learn an embedding model $\boldsymbol{\theta}$ for nodes in the graph of user $u_k$ and a predictor $\phi$ that can use the node embeddings to estimate the following probability:

$$\mathbb{P}\big((v_i, v_j, r) \in \mathcal{E}_k | \boldsymbol{\theta}, \boldsymbol{\phi}; \mathcal{G}_k\big). \tag{1}$$

Based on this probability we can then recommend new edges to the users. For each client, our rule recommendation model has two parts: a Graph Neural Network (GNN) to calculate the node embeddings and a predictor to predict the connections between nodes (the types of rules) as well as their probabilities. As discussed above, each client $u_k$ has a graph $\mathcal{G}_k$ to represent the ground truth connections between different nodes. We choose a two-layer graph neural network, GraphSage [10],

to get the embedding of nodes. After the two-layer GNN, we can then get the node embeddings $\mathbf{z}_v^k$ for all $v \in \mathcal{V}_k$. For each client $u_k$, after we get the node embeddings $\mathcal{Z}_k = \{\mathbf{z}_v^k, \forall v \in \mathcal{V}_k\}$. We then need to predict the edge connection probability $\mathbf{p}_{v_i,v_j}^k \in [0,1]^{|\mathcal{R}|}$ between node $v_i$ and node $v_j$ for all $v_i \in \mathcal{V}_k$ and all $v_j \in \mathcal{V}_k$. In our case, for the predictor model, we use a two-layer fully-connected neural network with ReLU activation for the first layer and the Sigmoid function in the last layer. The predictor uses $\mathbf{z}_{v_i}^k$ and $\mathbf{z}_{v_j}^k$, embeddings of node $v_i$ and node $v_j$ as input, and outputs the link probability of edges using its weights ($\phi$).

**Centralized Optimization**   The model can be easily trained in a centralized setting where the centralized server stores all user graphs $\mathcal{G}_k = (\mathcal{V}_k, \mathcal{X}_k, \mathcal{E}_k)$ for all $u_k \in \mathcal{U}$. We call this centralized training on the graph structure for rule recommendation as `GraphRule`. User graphs are mainly sparse, meaning most of connections between nodes are not set yet. Given the number of possible types of edges between nodes, there are only a few positive edges $\mathcal{E}_k^{\text{pos}}$. The other possible edges can then be considered as negative edges $\mathcal{E}_k^{\text{neg}}$. Considering all negative edges lengthens the training time, so we *sample negative edges* to balance the numbers of positive and negative edges.

We then use binary cross entropy loss including positive and negative edges as our objective function:

$$\mathcal{L}_k(\mathbb{P}^k, \mathcal{E}_k^{\text{pos}}, \mathcal{E}_k^{\text{neg}}) = -\frac{1}{|\mathcal{E}_k^{\text{pos}}| + |\mathcal{E}_k^{\text{neg}}|} \left( \sum_{e \in \mathcal{E}_k^{\text{pos}}} \log(p_e) + \sum_{e \in \mathcal{E}_k^{\text{neg}}} \log(1 - p_e) \right), \tag{2}$$

where $\mathbb{P}^k \in [0,1]^{|\mathcal{V}_k| \times |\mathcal{V}_k| \times |\mathcal{R}|}$ denotes all pairs of edge connection probability in $\mathcal{G}_k$, and $p_e \in [0,1]$ denotes the connection probability of a specific edge $e = (v_i, v_j, r)$.

## 2.2   FedRule

Learning the aforementioned models requires gathering the graph data of every user at a server to run the training. The `GraphRule` training, despite its fast convergence speed, can expose users' private data related to the devices or applications they are using and how they are connected together. Hence, it is important to facilitate a privacy-preserving training procedure to safeguard users' data. Federated learning [14] is the de-facto solution for such purpose in distributed training environments.

Recently, there have been some proposals to apply federated learning in recommendation problems such as in [20, 5, 15]. However, almost all these proposals are designed for cross-silo federated learning and not as granular as cross-device ones. In this paper, the problem of rule recommendation is formulated so as to be more suited for the cross-device federated learning setup. Although the cross-device federated learning setup is more desirable for the purpose of privacy-preserving algorithms, it makes the training procedure more challenging. The reason behind this is that for the problem of rule recommendation, the size of the data (*i.e.*, the graph structure for each client) is small and it follows non-IID distribution due to heterogeneous user behaviors. The non-IID data distribution increases the variance of gradients among users and makes the gradient updates coming from different users to go in different directions. Hence, the local training in the federated learning by averaging the gradients is hard to converge due to misaligned directions of gradients. As it is shown in Section 3, applying FedAvg with GNN on cross-device settings [14, 11] like rule recommendation problem can fail in some cases due to the non-IID problem mentioned above.

We then propose the `FedRule`, federated rule recommendation system with graph neural network. The design schema of the system is depicted in Figure 2. We use negative sampling, which samples negative edges in the graph, to balance the numbers of positive and negative edges. To avoid the non-IID problem, we use two control parameters in local machines for each client to correct the gradients and avoid drifting too much from the average model. `FedRule`, as presented in Algorithm 1 in the Appendix, consists of four main parts as follows:

**Local Updates**   At the beginning of each local training stage (communication round $c$), clients will get the updated global GNN ($\theta_k$) and prediction ($\phi_k$) models. Then, in each local iteration $t$, the client's device computes the gradients of models using local data. The gradient for the GNN part is with respect to the graph data ($\mathcal{G}_k$) and for the prediction model is with respect to the set of node embeddings ($\mathcal{Z}_k$) from the local graph. With the control parameters described next, the gradients get corrected and then the local models ($\theta_k^{(t,c)}, \phi_k^{(t,c)}$) get updated using their respective learning rates

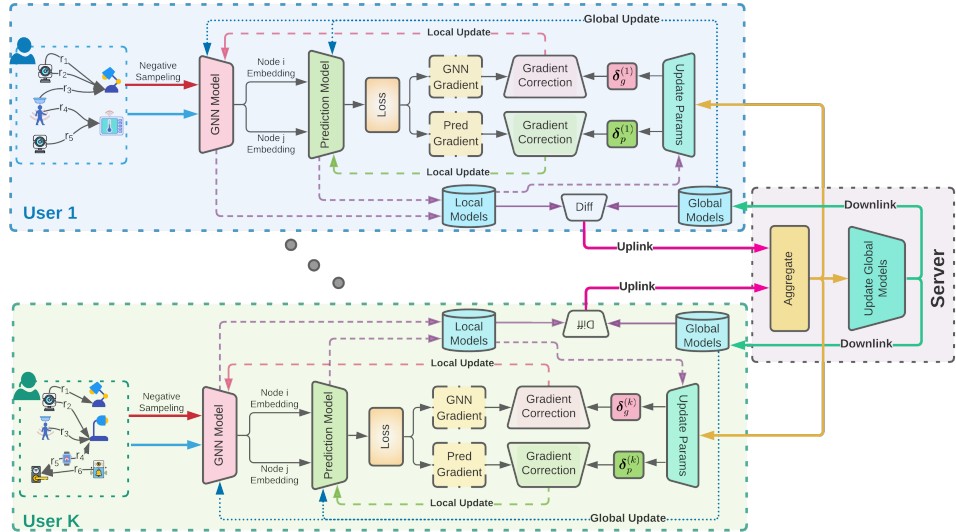

Figure 2: Overview of `FedRule` System Architecture Design.

$(\eta_\theta, \eta_\phi)$. Note that since the GNN model is a global representation of nodes and the prediction model is more of a personalized classifier, the learning rates of the models might be different.

**Gradient Correction**   For each local iteration, after computing the gradients, we adapt the FedGATE's variance reduction technique [9] and use two control parameters $(\delta_{\boldsymbol{\theta}_k}, \delta_{\boldsymbol{\phi}_k})$ for the GNN and predictor models respectively. The use of these control parameters will help the training to reduce the variance of convergence, as it can be inferred from the experimental results in Section 3 as well. Similar to learning rates, due to different natures of the models, they might get corrected with different rates using parameters $\lambda_{\boldsymbol{\theta}}, \lambda_{\boldsymbol{\phi}}$.

**Model Aggregation**   After $\tau_k$ local steps in each client's device, we aggregate the models from the devices. To do so, we first compute the difference between the current local models and the starting global models at round $c$, denoted by $\boldsymbol{\Delta}_{\boldsymbol{\theta}_k}^{(\tau_k, c)}, \boldsymbol{\Delta}_{\boldsymbol{\phi}_k}^{(\tau_k, c)}$. Then, the server averages over these updates from clients and send back these averages to the clients $(\boldsymbol{\Delta}_{\boldsymbol{\theta}}^{(c)}, \boldsymbol{\Delta}_{\boldsymbol{\phi}}^{(c)})$. Also, the server uses these average updates to update the global models that needs to broadcast to the clients in the next round. Secure gradient aggregation methods can be integrated in the system to better protect the privacy.

**Parameter Updates**   Using the calculated average updates in the previous stage, clients update their local control parameters using the deviation of the local updates from average updates:

$$\delta_{\boldsymbol{\theta}_k}^{(c+1)} = \delta_{\boldsymbol{\theta}_k}^{(c)} + \frac{1}{\eta_{\boldsymbol{\theta}} \tau_k} \left( \boldsymbol{\Delta}_{\boldsymbol{\theta}_k}^{(\tau_k, c)} - \boldsymbol{\Delta}_{\boldsymbol{\theta}}^{(c)} \right)$$
$$\delta_{\boldsymbol{\phi}_k}^{(c+1)} = \delta_{\boldsymbol{\phi}_k}^{(c)} + \frac{1}{\eta_{\boldsymbol{\phi}} \tau_k} \left( \boldsymbol{\Delta}_{\boldsymbol{\phi}_k}^{(\tau_k, c)} - \boldsymbol{\Delta}_{\boldsymbol{\phi}}^{(c)} \right) \tag{3}$$

## 3   Empirical Evaluation

### 3.1   Dataset

To experiment the efficacy of our proposed algorithm, we first use a real-world dataset for smart home devices. This proprietary dataset contains the rules that connect smart devices in different clients' houses. Hence, by nature the distribution of rules among different clients is non-IID, which is in line with the federated learning setting. We call this dataset the "*Smart Home Rules*" dataset, which contains $76,218$ users with $201,940$ rules. We simplify the current rules into the following form: $<$ trigger entity, trigger-action pairs, action entity$>$, where the trigger-action pairs denote the connection type. For instance, we can connect a smart doorbell to another camera, when by pressing

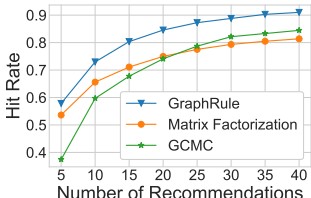 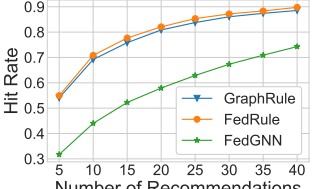 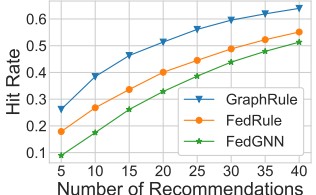

Figure 3: Hit Rate on the test set of Smart Home Rules Dataset vs. the number of recommendations, comparing `GraphRule` with conventional recommendation systems.

Figure 4: Hit Rate on the test set of Smart Home Rules Dataset vs. the number of recommendations, comparing `GraphRule` with federated approaches.

Figure 5: The Hit Rate on the test set for Smart Home Rules dataset, considering multiple entities with the same type. We compare `GraphRule` with `FedRule` and FedGNN.

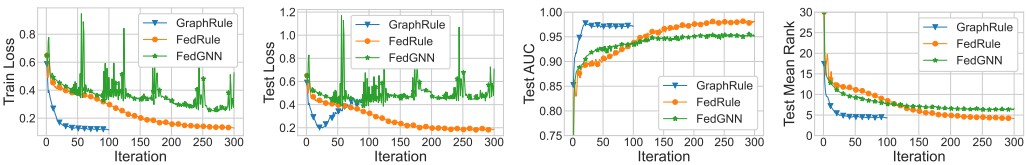

Figure 6: Train and test performance of different algorithms on Smart Home Rule Dataset. `FedRule` smoothly converges while FedGNN training diverges after a while.

the doorbell we want to power on the camera for recording. Then, the rule format is < Doorbell, Doorbell Pressed - Power On, Camera>. We have 11 unique entities and 163 unique trigger-action pairs, resulting in a total of $1,207$ unique rules. We also provide more experiments on the IFTTT dataset [6] in the Appendix. The details of experiments are deferred to the Appendix as well.

## 3.2 Comparison of Graph-Based and User-Item Based Methods

We first compare our centralized graph model, `GraphRule`, with user-item-based methods. For a fair comparison with user-item-based methods, we consider entities with the same type as one entity. As shown in Figure 3, the hit rate of `GraphRule` is outperforming both GCMC and Matrix Factorization in this task, which validates our analysis in Section 2.1. With the increase of the number of recommendations, `GraphRule` has a better hit rate, close to $0.91$, when recommends $40$ rules, which means by recommending $40$ rules among 1207 rules total, the user has $91\%$ chance to choose rules in the recommendation lists on average.

## 3.3 Evaluation of Centralized and Federated Algorithms

Now, we evaluate the performance of our proposed `FedRule`, and compare it with FedGNN training [20], as well as our proposed centralized `GraphRule`. We perform the evaluations on both train and test datasets. The results of applying the aforementioned algorithms on the Smart Home Rules dataset are depicted in Figure 6. During the training, the centralized algorithm, `GraphRule`, converges faster since it utilizes the training data of all users in an iid manner. For federated settings, given the heterogeneous data distribution among users and the heterogeneity of the number of rules and entities for each user, FedGNN converges to a higher loss with much higher variance. In the IFTTT dataset, FedGNN diverges after 50 iterations, as depicted in the Appendix. This is especially exacerbated by the large number of user with small amount of data (76, 218 users and 2.65 rules per user on average). On the other hand, `FedRule` smoothly converges to the same loss as the `GraphRule` due to its variance reduction mechanism. Similarly, the test AUC of `GraphRule` converges faster. During the training process, FedGNN has a better test AUC than `FedRule` at the start but increases slowly after 50 iterations as the training loss diverges. The test AUC of `FedRule` increases steadily to the same level as `GraphRule` after 200 iteration. The test Mean Rank also shows similar patterns. Also, Figure 4 shows the test hit rate of the Smart Home Rules dataset. `FedRule` and `GraphRule` have very close performance and are at most $26.8\%$ better than FedGNN.

### 3.4 Entities with Same Type

As it was mentioned, the graph structure allows us to distinguish between different entities with the same type in a user's graph. In this case, user-item-based methods become infeasible. User graph-based methods, however, can solve this problem by simply considering these entities as different nodes in the graph and using a node embedding to distinguish between different nodes with the same type. Figure 9 shows the training and testing results for centralized and federated methods. Similar to the previous part, `GraphRule` converges faster, and FedGNN diverges after 150 iterations. Similarly, the test loss of FedGNN diverges after 100 iterations while `FedRule` converges smoothly. Also, the AUC and Mean Rank results show that `FedRule` has a better performance than FedGNN. Moreover, Figure 5 shows test hit rates. `GraphRule` has a better performance than federated approaches.

## Acknowledgement

This research was partially supported by NSF grant CNS-1909306. The authors would like to acknowledge that the icons used in this paper are made by "Flaticon", "Justicon", "Nikita Golubev", and "Good Ware" from `www.flaticon.com`.

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

**Algorithm 1:** `FedRule` Federated Learning for Rule Recommendation Systems

---

**for** $c = 1, \ldots, C$ **do**

  **for** *each client* $k \in [K]$ **do in parallel**

    Set $\boldsymbol{\theta}_k^{(1,c)} = \boldsymbol{\theta}^{(c)}$, $\boldsymbol{\phi}_k^{(1,c)} = \boldsymbol{\phi}^{(c)}$,

    **for** $t = 1, \ldots, \tau_k$ **do**

      Set $\boldsymbol{g}_{\boldsymbol{\theta}_k}^{(t,c)} = \nabla_{\boldsymbol{\theta}_k} \mathcal{L}_k(\boldsymbol{\theta}_k^{(t,c)}, \boldsymbol{\phi}_k^{(t,c)}; \mathcal{G}_k)$

      Set $\boldsymbol{g}_{\boldsymbol{\phi}_k}^{(t,c)} = \nabla_{\boldsymbol{\phi}_k} \mathcal{L}_k(\boldsymbol{\theta}_k^{(t,c)}, \boldsymbol{\phi}_k^{(t,c)}; \mathcal{Z}_k)$

      `// Correct Gradients`

      $\tilde{\boldsymbol{g}}_{\{\boldsymbol{\theta}_k, \boldsymbol{\phi}_k\}}^{(t,c)} = \boldsymbol{g}_{\{\boldsymbol{\theta}_k, \boldsymbol{\phi}_k\}}^{(t,c)} - \lambda_{\{\boldsymbol{\theta}, \boldsymbol{\phi}\}} \delta_{\{\boldsymbol{\theta}_k, \boldsymbol{\phi}_k\}}$

      `// Update Parameters`

      $\boldsymbol{\theta}_k^{(t+1,c)} = \boldsymbol{\theta}_k^{(t,c)} - \eta_{\boldsymbol{\theta}} \, \tilde{\boldsymbol{g}}_{\boldsymbol{\theta}_k}^{(t,c)}$

      $\boldsymbol{\phi}_k^{(t+1,c)} = \boldsymbol{\phi}_k^{(t,c)} - \eta_{\boldsymbol{\phi}} \, \tilde{\boldsymbol{g}}_{\boldsymbol{\phi}_k}^{(t,c)}$

    **end**

    `// Update Control Parameters`

    send $\boldsymbol{\Delta}_{\{\boldsymbol{\theta}_k, \boldsymbol{\phi}_k\}}^{(\tau_k,c)} = \left\{ \boldsymbol{\theta}_k^{(c)}, \boldsymbol{\phi}_k^{(c)} \right\} - \left\{ \boldsymbol{\theta}_k^{(\tau_k,c)}, \boldsymbol{\phi}_k^{(\tau_k,c)} \right\}$ to server and gets $\boldsymbol{\Delta}_{\{\boldsymbol{\theta}, \boldsymbol{\phi}\}}^{(c)}$

    Update control parameters using Eq. (3)

  **end**

  `// Server Operations`

   `// Difference Aggregation`

  $\boldsymbol{\Delta}_{\{\boldsymbol{\theta}, \boldsymbol{\phi}\}}^{(c)} = \frac{1}{K} \sum_{k=1}^{K} \boldsymbol{\Delta}_{\{\boldsymbol{\theta}_k, \boldsymbol{\phi}_k\}}^{(\tau_k,c)}$ and broadcasts back to clients

   `// Update Global Models`

  Compute $\left\{ \boldsymbol{\theta}^{(c+1)}, \boldsymbol{\phi}^{(c+1)} \right\} = \left\{ \boldsymbol{\theta}^{(c)}, \boldsymbol{\phi}^{(c)} \right\} - \boldsymbol{\Delta}_{\{\boldsymbol{\theta}, \boldsymbol{\phi}\}}^{(c)}$ and broadcast to local clients

**end**

---

# A  `FedRule` Algorithm

The details of the `FedRule` algorithm can be seen in Algorithm 1.

# B  Additional Details on Experiments

## B.1  Experimental Setups

**Comparing Methods**  We compare these methods: (i) Matrix Factorization [3]: Complete user-item matrix by matrix factorization. GCMC [2]: Graph-based auto-encoder framework for matrix completion based on user-item bipartite graph. (ii) `GraphRule`: The proposed centralized optimization for graph formulation of rule recommendation. (iii) FedAvg with GNN [14, 11]: Each user has a user graph and a local model. The server aggregates the local models and uses FedAvg to train the global model. We call this FedGNN. (iv) `FedRule`: Our proposed federated rule recommendation algorithm.

**Experiment Setting**  We use the Adam optimizer with learning rate 0.1 and 100 training rounds. For federated algorithms, we use 3 local steps at each communication round, for 300 total iterations. The dimensions of the hidden states between the two GraphSage layers and the two NN layers are 16 and the number of possible trigger-action pairs, respectively. For `GraphRule`, users' graphs are stored in the central server and we do the gradient descent with all users' graphs. For federated methods, we compute the batch gradient descent on each user's graph with local epoch of 3 to train the local models, then the server aggregates the local models to update the global model. We set the hyper-parameter $\lambda = 1$ for FedRule algorithm. For each user, we use 80% rules set by the user for training and the remaining 20% rules for testing.

**Evaluation Metrics**  We use the following metrics to compare algorithms: (i) Loss: Binary cross entropy loss with the positive and negative edges. (ii) AUC: Area Under the Curve. (iii) Mean Rank:

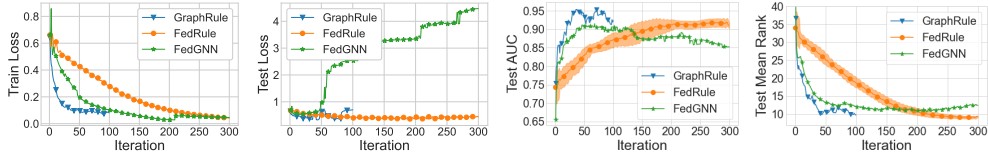

Figure 7: Training and test performance of different centralized and federated algorithms on IFTTT Dataset. Again, `FedRule` smoothly converges while FedGNN overfits.

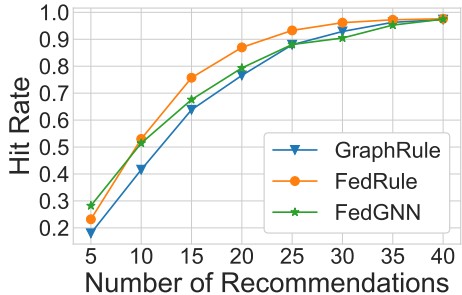

Figure 8: The Hit Rate on the IFTTT test dataset for centralized and federated algorithms with the rule filter.

Mean rank of positive testing edges between specific entities. (iv) Hit Rate@N: Recommend N rules and check if the positive test edges are included.

## B.2 Description of Datasets

**Proprietary Smart Home Rules Dataset**   As discussed in the main body, we primarily use this dataset for our comparisons, which consist the data of smart home devices' rules. The types of entities and trigger-action pairs are shown in Table 1.

**IFTTT Dataset**   The IFTTT Dataset [6] is one of the most popular EUD tools. To the best of our knowledge, this is the only publicly available dataset of IF-THEN rules defined and shared by different users. But it does not support entities with the same type. It was obtained by Ur et al. [17] with a web scrape of the IFTTT platform as of September 2016. The dataset contains $144$ different users with $8,729$ rules. There are $3,020$ types of rules used by users. In the current dataset, we use $53$ unique entities and $132$ unique trigger-action pairs. The types of entities and trigger-action pairs are shown in Table 2.

| Types of Entities | Types of Trigger-Action Pairs |
|---|---|
| Camera | Open, Power On |
| Chime Sensor | Open, Power Off |
| Contact Sensor | Open, Motion Alarm On |
| Light | Open, Change Brightness |
| Lock | ... |
| Mesh Light | Open, Siren On |
| Motion Sensor | Open, Alarm Action |
| Outdoor Plug | ... |
| Plug | Person Detected, Power Off |
| Thermostat | Smoke Detected, Power Off |
| Outdoor Camera | Doorbell Pressed, Power On |

Table 1: Smart Home Rules Dataset entities and trigger-action pairs.

| Types of Entities | Types of Trigger-Action Pairs |
|---|---|
| Android Device | New Post, Share a Link |
| Weather | New Follower, Post a Tweet |
| Gmail | New Like, Add File from Url |
| YouTube | New Liked Video, Create a Post |
| ... | ... |
| Facebook | New Photo, Send Me an Email |
| Instagram | New Photo, Add File from Url |
| Linkedin | New Screenshot, Add File from Url |
| ... | ... |
| Twitter | You Exit an Area, Set Temperature |
| Reddit | Battery Low, Send an Sms |

Table 2: IFTTT Dataset entities and trigger-action pairs.

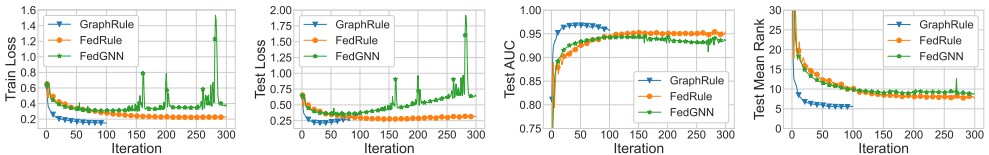

Figure 9: Training and test performance of different centralized and federated algorithms on Smart Home Rule Dataset, when considering multiple entities with the same type.

|            | Loss   | AUC    | MR    | MR(RT) |
|------------|--------|--------|-------|--------|
| GraphRule  | 0.1997 | 0.9768 | 4.349 | 3.0970 |
| FedGNN     | 0.3878 | 0.9521 | 6.661 | 5.3946 |
| FedRule    | **0.1892** | **0.9804** | **4.156** | **2.9150** |

Table 3: Final test results on Smart Home Rules Dataset (RT: remove rules shown in training graph).

|            | Loss   | AUC    | MR      | MR(RT)  |
|------------|--------|--------|---------|---------|
| GraphRule  | **0.3238** | **0.9491** | 9.6997  | 9.6745  |
| FedGNN     | 0.4905 | 0.9072 | 12.4479 | 12.4240 |
| FedRule    | 0.3614 | 0.9417 | **8.8812** | **8.8564** |

Table 4: Final test results on IFTTT Dataset. (RT: remove rules shown in training graph).

## B.3 Additional Evaluation of Centralized and Federated Algorithms

**Additional Results on Smart Home Dataset**    Table 3 shows the final test results of Smart Home Rules data. Although the `GraphRule` has access to all users' graphs during the training, `FedRule` slightly outperforms `GraphRule` since it employs variance reduction, making the convergence more smooth. `FedRule` greatly outperforms FedGNN in all criteria. The mean rank of `FedRule` is $2.915$ after removing the rules in training graphs, which means that for any rules between two specific entities, we need to recommend three rules in average and the user is very likely to adopt at least one of them.

**Results on IFTTT Dataset**    The results of applying these methods on IFTTT dataset is shown in Figure 7. The IFTTT dataset is relatively smaller than the Smart Home Rules data, so it is easier to converge during the training process. For the training loss, the centralized training, `GraphRule`, convergences faster since it benefits from iid distribution of data. For federated setting, given it is a small dataset, the train loss of FedGNN converges faster than `FedRule` with the same learning rate but the test loss of FedGNN diverges after $50$ iterations. Similar to the Smart Home Rules dataset, FedGNN has a better test AUC at the start but it decreases after $50$ iterations as the test loss diverges. The test AUC of `FedRule` increases steadily during the training. Also, its test mean rank smoothly decreases, converging to even lower value than the `GraphRule`. Table 4 shows the final test results on IFTTT data. Although `GraphRule` has a better testing loss and a slightly better AUC, the Mean Ranks of `FedRule` are better than those of `GraphRule`, which is a better indicator of the recommendation performance.

Figure 8 shows the hit rate of different algorithms in IFTTT dataset. Given the huge number of rules and apps, there are trigger-action pairs that are infeasible between some entities in practice. Since the number of users are small and hard to train a general model to avoid these infeasible pairs, we do rule filtering on the recommended rules to keep valid rules. Due to the small size of the dataset, the evaluation has high variance. FedGNN is slightly better than GraphRule and FedRule for top-5 recommendation. But `FedRule` is the best in general and has at the best $8.1\%$ higher hit rate than `GraphRule` and FedGNN.

