# OpenReview forum: "FedRule: Federated Rule Recommendation System with Graph Neural Networks"
_NeurIPS.cc/2022/Workshop/Federated_Learning — FL-NeurIPS 2022 Poster_

### Official Review · Reviewer_j4yc · 2022-10-11
**Federated Rule Recommendation**

The paper considers a specific instance of recommendation systems in a federated setting. Rule recommendation is the problem of suggesting rules between entities to users that may then adopt those rules, as illustrated in the IoT/smart home examples. Since training data is collected based on rules used in existing smart homes, it is considered private information and should not be shared. This motivates a federated learning framework where raw rule data is not centralized for training. The FedRule method replaces "User-Item Matrix" recommender systems with a graph-based system, where each entity is a node and each rule is an edge. FedRule learns the graph structure as embeddings via a GNN, and predicts the existence of edges with a predictor model. The embeddings and predictors must be learned in a federated system with update averaging.

Quality:
The paper contains several typos (e.g. L9-10, L36, L70, L77-78, L120-122, L191), and it omits the descriptions of key elements (variance reduction, baseline methods).
It appears that the test loss increases in Fig 6b for GraphRule showing some form of model collapse or overfitting which is not addressed.

Clarity:
The proposed method is clear, but the justification in Sec 2.1 is not entirely convincing. There are several references to techniques or datasets which are not described (e.g. GraphSage, GCMC, IFTTT) making the comparisons difficult to follow. The variance reduction technique in particular is touted as a crucial component for enabling the system to learn in non-IID settings, but was not explained.

Originality:
The idea of using federated GNNs for recommendation is not novel. The authors try to draw a distinction between cross-silo (previous methods) and cross-device (this method) settings, however, I am not convinced that this is a significant distinction. The main idea here seems to be the use of variance reduction techniques to better facilitate learning on small non-IID datasets, but the techniques are not described in detail. Previous methods employ better systems for privacy-preservation [Wu et al. "FedGNN: Federated graph neural network for privacy-preserving recommendation"].

Significance:
FedRule appears to bring in previously used variance reduction techniques (FedGATE) to the setting of FedGNN, which limits its overall significance.

Pros:
 - Rule recommendation will increase the usefulness of IoT devices and smart home networks.
 - FedRule addresses a limitation of User-Item Matrix approaches - there may be multiple instances of the same entity with different roles.
 - The performance of FedRule appears better than FedGNN in the cross-device setting.

Cons:
 - All experiments in the main text use a proprietary dataset making the results unverifiable.
 - Sharing data on entities and rules used in a smart home is a major privacy and security concern. It may even be a personal safety concern if a home's security system is included in the data. The authors state that centralized FL is the "de facto solution", but it is not sufficient to address the privacy and security concerns of this setting. The authors should discuss the limitations and risks of FL in terms of privacy and propose additional measures to guarantee personal data cannot be extracted from the gradients that are shared.
 - FedRule provides a much weaker privacy guarantee than FedGNN.

---

> ### Author Response · Authors · 2022-10-24
> **Discussion on privacy and security concerns**
>
> For cons 2 and 3:
> In [Wu et al. "FedGNN: Federated graph neural network for privacy-preserving recommendation"], it is cross-silo and directly shares the node features, which is a serious privacy leakage. It adds differential privacy (random noise on the node feature) to provide a sort of "privacy". Our method is cross-device and only shares the gradients. Our method provides a much better privacy guarantee than FedGNN (Wu et al. ).
>
> There are always concerns about gradient sharing for all FL papers, which is also a good suggestion especially since our algorithm will be deployed in the real system. Our system can integrate any privacy-preservation method for gradient sharing. I am sure I will mention secure gradient sharing in every FL-related paper. :)
>
> PS: The typo issue is quite strange. The original version is good but the submitted version is missing words. Thank you very much for mentioning this!

---

### Official Review · Reviewer_6Bx9 · 2022-10-13

The paper proposes a rule recommendation systems for the IoT devices based on the federated learning with graph neural networks. Specifically, instead of user-item recommendation problem, the paper re-formalize the problem into a link prediction problem in the constructed device-device graph and uses the federated learning to incorporate different users' features to solve data insufficiency problem. The experiments conducted compares the proposed method with matrix factorization method and show it could achieve a better performance.

Pros:
1. The idea to formalize the rule recommendation problem into graph one is novel and interesting.
2. The proposed federated learning framework could effectively solve the data insufficiency problem.
3. The experiment results show the effectiveness of the proposed method.

Cons:
1. The method section is not very clear. It would be better to define the GNN properly, i.e. the \theta_k and \phi_k. Also, it is not obvious about the difference of global model and local model. There are some settings in FL that \theta would be shared and \phi is not shared.  At the same time, if the \theta_k and \phi_k could be thought as the model parameters, it seems not necessary to explicitly define those since the updating rule is exactly the same.
2. As the proposed method uses FedGATE’s variance reduction technique in the aggregation, I am not sure the main novelty of the proposed methods FedRule compared with FedGATE. And it is not clear why it could help to solve the heterogeneous and sparse data problem.

---

### Official Review · Reviewer_JZ5M · 2022-10-14
**Interesting idea with thorough experimentation**

In this paper the authors propose a privacy-preserving rule recommendation system for smart homes. The main insight is to cast the rule recommendation problem as a link prediction task, which allows one to perform Federated Learning with GNNs. However, straight-forward application of FL would not work since the domain impose restrictions on the size of local data available to each node and a non-ID regime. This is tackled through a novel gradient control scheme at each node, inspired by (Haddadpour et al., 2021).

The authors present three main contributions. In my opinion, the first one is fair, but not that strong; the second one is a good contribution; and the third one is not a contribution, but the empirical support itself for the solution proposed. I recommend the authors rephrase these contributions and put the third contribution as an extra paragraph relating to empirical support.

The paper is well written, and experimentations are well done. Comparison with main contender (FedGNN) seems to be unfair, since it was not built for this, but I understand that there might not be a better method to compare against.

Overall, I believe the paper presents an interesting application of controlling for variance locally in the domain of FL with GNNs as applied to cross-devices (more specific).

---

### Decision · Program_Chairs · 2022-10-20

Accept (Poster)